# Association between Dietary Inflammatory Index and Bone Mineral Density Changes among Pregnant Women: A Prospective Study in China

**DOI:** 10.3390/nu16030455

**Published:** 2024-02-05

**Authors:** Xiaoyu Zhu, Yalin Zhou, Zhang Wen, Wanyun Ye, Lan Gao, Yajun Xu

**Affiliations:** 1Department of Nutrition and Food Hygiene, School of Public Health, Peking University, No. 38 Xueyuan 7 Road, Beijing 100191, China; xiaoyuzhu@bjmu.edu.cn (X.Z.); 2216393053@bjmu.edu.cn (Y.Z.); 1710306240@pku.edu.cn (Z.W.); yewanyun_vera@bjmu.edu.cn (W.Y.); 2Beifang Branch of Peking University Third Hospital, Chedaogou No. 10, Beijing 100089, China; 2211110219@stu.pku.edu.cn

**Keywords:** dietary inflammatory index, serum inflammation factors, bone mineral density, Chinese pregnant women

## Abstract

Objectives: This study aims to examine the relationship between dietary inflammatory index (DII) and bone mineral density (BMD) changes among Chinese pregnant women, offering valuable insights for dietary guidance during pregnancy. Methods: 289 pregnant women were enrolled in this cohort. Serum inflammatory factors and ultrasonic BMD were measured at the first, second, and the third trimesters. DII scores were calculated based on a semi-quantitative food frequency questionnaire (FFQ) and divided into tertiles. We compared the differences in inflammatory factors in serum across the tertiles of DII and changes in BMD at the second and third trimesters across the tertiles. Results: The participants with higher DII scores had higher total energy intakes than those with lower DII scores. The serum level of interleukin-6 (IL-6) was significantly different across the tertiles of the DII. Women who had lower DII scores had higher T-scores and Z-scores in the BMD assessment. In the test of trends, after adjusting potential covariates, including educational level, physical activity, body mass index, and calcium, vitamin D, or multivitamin supplements, DII values were determined to be positively related to the maternal BMD lost. Conclusions: DII was positively associated with serum IL-6. Meanwhile, higher DII scores were associated with more bone mass loss in pregnant women. We recommend adhering to a lower-DII diet to preserve BMD during pregnancy.

## 1. Introduction

Diet plays a crucial role in regulating the body’s inflammation levels through pro- and anti-inflammatory mechanisms. Recently, there has been widespread reporting on the concepts of pro-inflammatory and anti-inflammatory diets [1,2]. Diets characterized by high intakes of red meats, vegetable oil, refined carbohydrates, processed meat, soft drinks, or artificial trans fats were defined as pro-inflammatory diets, for their food groups could activate inflammatory responses [3,4]. Diets rich in fruits, vegetables, whole grains, fish, or green tea have been associated with reduced inflammation, categorizing them as anti-inflammatory [5,6,7]. To date, various dietary indices have been reported, each attempting to evaluate overall dietary quality. Among them, the dietary inflammatory index (DII) has been widely used in assessing the relationship between dietary habits and inflammation and in predicting the inflammatory potential of an individual’s diet, including with regard to C-reactive protein (CRP), interleukin-6 (IL-6), and tumor necrosis factor (TNF)-α levels [8]. The higher the DII score, the more pro-inflammatory the diet and the higher the possible inflammatory potential [9,10,11,12]. A high DII has been proved to be associated with diseases such as cancer, obesity, and type 2 diabetes mellitus. For example, DII scores are inversely associated with skeletal muscle mass among boys [13] and positively associated with the prevalence of hyperglycemia among men and the prevalence of central obesity among postmenopausal women [14]. Regarding the effect of a pro-inflammatory diet on BMD, it has also been reported that a high DII score may be a risk factor for lower BMD in the lumbar spine among postmenopausal Iranian women [15].

Inflammation has also been linked to adverse maternal and infant outcomes; thus, the DII is now a common index for assessing the risks posed by the diets of pregnant women [16]. Additionally, the adherence to a pro-inflammatory diet during pregnancy is associated with maternal systemic inflammation and may be associated with impaired fetal growth and breastfeeding failure [17]. 

Chronic inflammation is associated with several diseases [18]. CRP and IL-6 were used to assess the association between inflammation and diseases such as cardiovascular disease, preeclampsia osteoporosis, and even bone mineral density [19,20]. Furthermore, in experimental studies, inflammation factors have also been reported to be associated with the alteration of bone structure [21]. Thus, the circulating levels of inflammatory markers might predict changes in BMD and resorption in adults [22,23,24,25].

Calcium consumption is essential for bone development and maintenance throughout one’s life. During pregnancy, an increasing pressure for mineral intake is placed on the mother due to the calcium demands of her fetus, and the maximal fetal demand for calcium occurs during the third trimester [25,26,27,28]. However, the adjustments to maternal calcium homeostasis begin in early pregnancy. Thus, calcium intake is especially crucial during pregnancy and lactation, and there are potential adverse effects on maternal bone health if maternal calcium stores are depleted. Several studies have tried to quantify pregnancy-induced bone loss and found permanent alterations in the skeletal structure with implications for the metabolic and mechanical function of bones [26]. Pregnancy and lactation might be a risk factors for developing osteoporosis [29]. The DII was also reported to be associated with BMD. For example, Woolford found that during late pregnancy, a higher pro-inflammatory diet is negatively associated with offspring bone measures, supporting the importance of maternal and childhood diets for longitudinal offspring bone health [30]. However, the relationship between DII and BMD has been less explored, especially among pregnant women.

Thus, we aim to evaluate the relationship between DII and BMD during the pregnancy period. Furthermore, serum chronic inflammation indicators were measured, and their association with DII scores was also investigated. Our goal is to explore the dietary risk factors of BMD, providing nutrition recommendations to improve the well-being of pregnant women.

## 2. Materials and Methods

### 2.1. Study Design and Participants

The primary objective of this study was to explore the interplay between diet and bone density changes among pregnant women during the second and third trimesters. Thus, participants were informed of the purpose of the inquiry by trained researchers. Basic information about the study was given, and individual informed consent was collected.

A cohort was set up in the Changping Maternity and Child Care Hospital from 1 November 2020 to 1 November 2021, in Beijing, China. Apparently healthy pregnant women at the first trimester were recruited, and those with acute or chronic diseases; having difficulty communicating; afflicted with serious diseases that affect bone metabolism, including a history of thyroid problems, renal failure, malignancy, or rheumatoid arthritis; or who had undergone hormone replacement therapies were excluded. The participants who completed the quantitative Food Frequency Questionnaire (FFQ) and participated in ultrasonic BMD examinations at the first, second, and third trimesters were included with the total number of 289 participants. Written informed consent was provided by all participants, and the study proposal was approved by the committee on Medical Ethics of Peking University.

### 2.2. BMD Measurement

The patients’ calcaneal BMD values were measured at the second and third trimesters by well-trained technicians using a Quantitative Ultrasound System employing a CM-200 device (Furuno Electric, Nishinomiya City, Japan), which was calibrated using the matched module prior to each measurement, with a variation coefficient of 0.19%. The speed of the ultrasonic wave propagation (SOS), Z-scores, and T-scores were measured. The Z-score indicates the standard difference between the SOS of the subjects and the average value of the age-matched population. The T-value is the standard difference between the SOS of the subjects and the average of the young-adult population.

### 2.3. Assessment of Inflammatory Factors of Blood Samples

Venous blood samples were taken from the participants during the second and third trimesters. Serum samples were collected after a rapid centrifugation of the blood samples. The levels of inflammatory factors including CRP (C-reactionprotein), IL-6 (Interleukin-6), IL-10 (Interleukin-10), IL-4 (Interleukin-4), and IL-1β (Interleukin-1β) were measured at Shanghai Crystal Day Biotech Co., Ltd. (Shanghai, China).

### 2.4. Measurement of Bone Turnover Markers

Serum osteocalcin (OC), osteoprotegerin (OPG), parathyroid hormone (PTH), nuclear factor κB receptor activating factor ligand (RANKL), and *N*-terminal propeptide of type I procollagen (PINP) levels were determined using an Elisa kit. The intra-assay coefficients of variation were less than 8%. All the kits were provided by Beijing Zhongshangboao Biological Company (Beijing, China).

### 2.5. Dietary Assessment

A semi-quantitative FFQ was designed based on a previously validated FFQ with some modifications [31]. In brief, the participating women were asked to report their intake frequency and amount of each food item in the FFQ, including refined cereals, whole cereals, soy and soy products, dark leafy vegetables, light vegetables, dark fruits, light fruits, poultry, meat and processed meats, fish and aquatic products, eggs, milk and dairy products, fungi and algae, nuts, baked bread, candies, junk food, fruit and vegetable juice, and soft drinks. The daily intakes of each item were calculated by multiplying the frequency and amount. For each item in the questionnaire, a standard reference quantity is given, which was used in conjunction with the food standard model and the retrospective dietary survey supplementary reference food atlas, with the size of the “hand” used as a reference, to help the subjects better estimate their daily intake. The daily total energy and nutrient intakes were calculated based on the Chinese Food Composition Table [32]. Calculations of daily flavonoid, proanthocyanidin, vitamin D, and calcium (from food and dietary supplements) consumption were conducted by referring to the US Department of Agriculture database. The isoflavones were examined with reference to the Hong Kong database.

### 2.6. DII Calculation 

The DII consists of 45 parameters on foods, nutrients, and other bio-active components. The Z score of each item was calculated using the following equation: Z score = (daily intake − mean)/standard deviation (SD) of each item. The mean and SD were derived after linking to the regionally representative world database [33]. To minimize the effects of outliners or non-symmetrical distributions, the Z scores were subsequently converted to a centered percentile score [15]. Then, they were multiplied by the inflammatory effect index of each parameter to get the DII_i_. The products of each parameter were then summed to attain the DII of each participant. 

### 2.7. Covariates

Covariates included body mass index (BMI (kg/m^2^)), education level, physical activity, and calcium, vitamin D, or multivitamin supplements [34,35,36]. BMI was calculated by referring to the weight/height^2^ values recorded in medical records at the second and third trimesters. Educational status was classified as high school or below, college, or postgraduate or above. Physical activity levels were represented as the total hours per week based on a 7-day total activity recall [37].

### 2.8. Statistical Analysis

All data analysis using SPSS 24.0. A *p* value < 0.05 indicates a significant statistical significance. All participants were divided into three groups according to the tertiles of the DII. The differences between the continuous and categorical variables among the groups were tested using one-way ANOVA and the Chi-square test. A test for linear trends was conducted by including the median value of each DII tertile as a continuous variable in the multiple linear regression models. For multiple linear analyses, two models were established for adjusting covariates: Model 1, for which there was no adjustment, and Model 2, a multivariate analysis, which was adjusted for demographic characteristics, namely, age, BMI, baseline T-score, METs, educational levels, physical activity, daily energy intake, time spent on physical activity, and calcium, vitamin D, or multivitamin supplements.

## 3. Results

### 3.1. Demographic Characteristics of Participants across Tertiles of DII

The mean of the DII scores was 59.69 ± 54.66 (mean ± SD). The lowest DII score was −76.05, and the highest was 246.68, indicating that the DII values are very different between different individuals’ diets. We calculated the tertile of DII and divided all the participants into groups G1(7.13 ± 15.97), G2(53.05 ± 11.6), and G3(119.53 ± 45.0) ordered from the lowest to the highest DII scores. The demographic characteristics of the three groups are shown in Table 1. There were no significant differences among the three groups in terms of age, BMI, education level, or physical activity. Regarding nutrient supplements, no differences were observed in terms of calcium and VD intakes but isoflavone is different across tertiles (*p* < 0.001). Furthermore, no differences were observed in the time spent in the sunshine across the three groups.

### 3.2. Distribution of Food Groups across Tertiles of DII

A total of 20 food groups were included in the FFQ, and their intakes across the tertiles of the DII are shown in Table 2. The ones in the highest DII tertile who consumed the most pro-inflammatory diet had the highest daily energy intake (2310 kcal/day). Those in the lowest DII tertile who consumed the most anti-inflammatory diets had the lowest daily energy intake (1957 kcal/day). For refined cereals, whole cereals, soy and soy products, dark leafy vegetables, light vegetables, light fruits, dark fruits, poultry, meat and processed meats, eggs, milk and dairy products, fungi and algae, and nuts, the intake increased when moving from groups G1 to G3 (*p* < 0.05). Furthermore, a similar level across three groups for fish and aquatic products, baked bread, candies, junk food, fruit and vegetable juice, and soft drinks (*p* > 0.05).

### 3.3. The Serum Inflammation Factor Level across the Tertiles of DII

Since DII has been widely reported to be associated with inflammation, we compared the participants’ serum levels of anti-inflammation and pro-inflammation factors. The differences in serum CRP, IL-6, IL-10, IL-4, and IL-1β levels across the three groups are shown in Table 3. A significant difference existed in the serum levels of IL-6 across the tertiles of DII, and its levels increased moving from groups G1 to G3. The levels of CRP and IL-1β also increased with a higher DII, but not statistically significantly (*p* > 0.05). Furthermore, a slight decreasing trend can be observed in IL-10 and IL-4 levels with DII, but this is also nonsignificant.

To further investigate the relationship between DII and serum level of inflammation factors, CRP, IL-6, IL-10, IL-4, and IL-1β were measured, and their associations with DII score were tested in the third trimester. The results show that the DII score was significantly related to IL-6 (*p* < 0.05) in the third trimester, while the other inflammation factors, including CRP, IL-10, IL-4, and IL-1β, were not significantly related to the DII score (*p* > 0.05) (Figure 1).

### 3.4. Maternal BMD Levels of DII Score Groups

Then, we analyzed the bone metabolism of the participants across the DII tertiles. As shown, the RANKL level was significantly different across DII (*p* = 0.002) (Table 4). Although the levels of OC, OPG, PINP, and PTH were not significantly different across DII scores, a decreasing trend was found with the increasing DII (*p* > 0.05). Meanwhile, the ratio of RANKL/OPG in G3 was 0.017, which was the highest, and the lowest ratio was 0.0085 in G1. These data indicate that the DII score is associated with pro-inflammation factors, especially IL-6, affecting bone metabolism. 

Next, we analyzed the SOS, Z-scores, and T-scores of the participants. As shown in Table 5, the average Z-scores of the participants were −0.44, −0.25, and −0.11, while the T-scores were −0.62, −0.28, and −0.09 in G3, G2, and G1. We found that the higher the DII score, the lower the T-score and Z-score. Additionally, there was a significant difference in the T-scores across the tertiles of DII for the participants in the third trimester (*p* < 0.05).

We also analyzed the T-scores of the participants during the entire pregnancy period across the tertiles of DII. As shown in Figure 2, the T-scores are relatively higher in the first trimester compared to those of the second and third trimesters in all three tertiles. Meanwhile, the T-scores of the participants in all trimesters across G1 tertiles were higher than those in the G2 and G3 tertiles, further indicating that DII score is associated with the T-score during the entire pregnancy period.

### 3.5. The Linear Association between DII and BMD

To further understand the relationship between DII and BMD, we analyzed the linear trend (based on T-score) using multivariable adjusted linear regression models. Model 1 is crude, and the adjusted means of BMD are shown in Table 6. After adjustment by demographic characteristics such as educational level, physical activity, time spent engaging in physical activity and in the sunshine, and calcium, vitamin D, or multivitamin supplements, a significant linear trend between DII and BMD was observed in Model 2. Women in G1 group had less bone mass loss than those in G2 and G3. 

## 4. Discussion

An effective dietary approach can prevent non-communicable diseases. To evaluate a diet, we normally focus more on the quality of food intake as a percentage but not an individual component since the relationship between foods is complex and as interactions exist between different foods. Assessing overall diet could thus help us to understand the interrelationships between dietary components [38]. Here, we included the DII score to assess the relationship between diet and inflammation and predict the inflammatory potential of an individual’s diet [33]. 

It was reported that a higher DII score indicates a more pro-inflammatory diet and a higher risk of developing an inflammatory potential [11,12,13]. Inflammation during pregnancy has been linked to adverse maternal and infant outcomes. Thus, we studied the diets of Chinese pregnant women based on the FFQ. A higher DII score was associated with a higher total energy intake, indicating a different dietary intake with a lower DII score. Higher DII scores were associated with higher IL-6 levels. Meanwhile, we also reported that the values of the bone metabolism item RANKL were different across DII tertiles, indicating that bone metabolism could be a marker of the BMD of pregnant women, and that DII score is negatively associated with BMD.

Regarding dietary intake, we found that a pro-inflammatory diet, as reflected in higher DII scores, was associated with a higher daily energy intake. Moreover, regarding the DII scores, the intakes of refined cereals, whole cereals, soy and soy products, dark leafy vegetables, light vegetables, light fruits, dark fruits, poultry, meat and processed meats, milk and dairy products, fungi and algae, and nuts were all higher, while the intake of fish and aquatic products, baked bread, candies, junk food, fruit and vegetable juice, and soft drinks did not influence DII-associated dietary habits. The parameters of the DII mostly stemmed from certain food groups, such as refined cereal, which accounted for the largest proportion of the diets and might have led to the significant difference across the DII groups. The consumption of candies or junk foods indicated a greater energy intake but affected the scores less. We hypothesize that this may because the overall proportion of these foods is small compared to that of refined cereal. 

In our study, we found that the serum levels of IL-6 were significantly different across the tertiles of the DII, and this result is consistent with other studies, revealing that inflammatory diets were associated with higher serum levels of inflammatory factor. The higher circulating levels of inflammatory markers also predict bone loss and resorption. For example, IL-6 is an inflammatory cytokine that is associated with bone loss and resorption. High levels of IL-6 were also reported to induce bone loss in individuals with osteoarthritis. Meanwhile, IL-6 and CRP inhibit the functions of osteoblasts, including the promotion of the proliferation, differentiation, and activation of osteoclasts, causing bone erosion and bone mass loss [39,40,41]. Thus, people consuming diets with a high DII value may suffer more inflammatory and bone loss problems. Reduced serum inflammatory levels were shown to be conducive to improving BMD [42]. 

BMD changes throughout a trimester due to the high calcium demand from the growth of the fetus and changes in maternal hormones [43]. Decreased BMD has been more widely reported in pregnant women compared with non-pregnant women. Pregnancy can cause reversible bone loss, especially at trabecular sites [29,44]. DII has widely been reported to be associated with BMD in adults and older individuals, but there are currently few studies that have evaluated the association between the DII and bone metabolism and density among pregnant women in China. In our study, BMD was measured in participants using an ultrasonic method at the second and third trimesters. We found that participants with a higher DII score along with a higher serum level of IL-6 showed a high level of RANKL and a higher ratio of RANKL/OPG, indicating a risk of more bone mass loss. RANKL has been widely reported to be associated with bone mass loss. For example, Hu et al. showed that RANKL promotes osteoclast formation and bone loss [45]. RANKL can be considered an early marker of bone mass loss for pregnant women, and inflammatory factors can interact with osteoblasts and osteoclasts to change the expression levels of RANK and RANKL, thus affecting BMD [46,47]. Interventions targeting inflammatory levels can directly impact bone health. Here, we connected the association between RANKL, bone mass loss, inflammatory factors, and DII scores among pregnant women. A low-DII diet may lower the risk of bone loss among pregnant women. It needs to be pointed out that the consumption of soybean in G3 was higher than that in G1, but the maternal BMD level in G3 was lower than that in G1. Although it was known that soybean intake reduced inflammation levels and inhibited bone loss due to estrogenic isoflavones, the function here was not as potent as that observed in previous reports [48,49]. This may be due to the relatively low intake of isoflavone here, which was suggested to prevent BMD loss around 80 mg/d in postmenopausal women on a previous study [50]. The intrinsic estradiol plasma levels are huge in pregnant women during the 2nd and 3rd trimesters; thus, the low affinity of isoflavones on the estradiol receptors may also affect the function of isoflavones on the prevention of BMD. Meanwhile, the health benefits of isoflavones are very depend on the equol; the ability of an individual to produce equol is very different which could also affect the benefit of isoflavones. The small sample size and single-center limitation in this study could be another factor that prevented us from observing a correlation between the intake volume of isoflavone and BMD. 

After adjusting for potential covariates of the multivariate linear model, the present study showed a significant linear relationship between DII and maternal BMD from the second to third trimester of pregnancy. The DII score used in our study, which estimates the pro-inflammatory effect of diet, may be a risk factor for lower BMD among pregnant women [51]. It also reminds us to pay attention to bone changes in young, pregnant women, which may increase the risk of osteoporosis later in life [52].

The present study, to the best of our knowledge, is the first to evaluate the association between BMD change and DII among Chinese pregnant women. However, it also has several limitations. First, the dietary data were collected based on a validated FFQ, in which recall bias could not be avoided, and self-reported dietary data have been known to suffer from measurement errors. Second, the use of ultrasonography for BMD measurement may have limitations compared to more precise methods such as dual-energy X-ray absorptiometry. Third, only a few confounding factors were adjusted in the linear models, but other potential factors may also exist. Fourth, the cohort study design allows for the identification of associations, but it cannot establish causation, and more factors that could contribute to the DII and BMD need to be explored. Fifth, the study sample size was small, and it was only a single-center study, so larger cohorts and multicenter studies are necessary. 

## 5. Conclusions

The results of the current study could provide some basic information on BMD evolution during pregnancy through the analysis of maternal diet. In the present study, the pro-inflammatory diet identified by a higher DII score was correlated with a higher daily energy intake. Moreover, a positive relationship was observed between higher DII scores and higher serum IL-6 levels. Consistently, higher DII scores were associated with higher bone mass loss in pregnant women. This study suggests adhering to a low-DII diet to maintain BMD during pregnancy. Reasonable control of total energy intake and the adjustment of diet composition may be good choices for pregnant women in order to reduce bone mass loss and prevent later osteoporosis. Further research is warranted to confirm these findings among more diverse populations.

## Figures and Tables

**Figure 1 nutrients-16-00455-f001:**
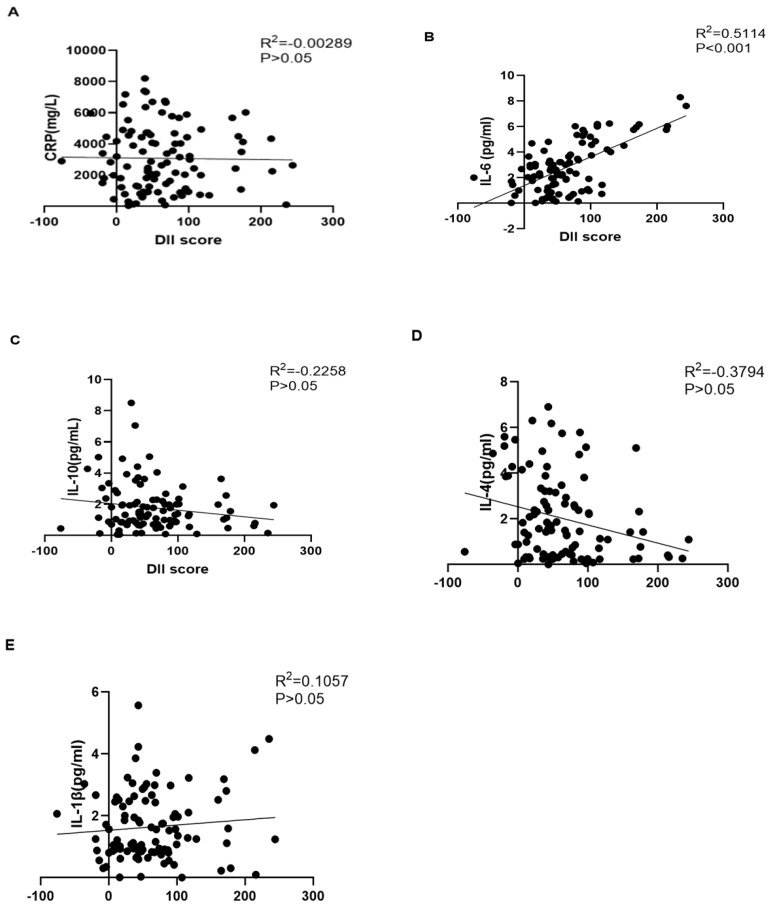
The relationship between DII and CRP (**A**), IL-6 (**B**), IL-10 (**C**), IL-4 (**D**), and IL-β (**E**) among 289 pregnant women.

**Figure 2 nutrients-16-00455-f002:**
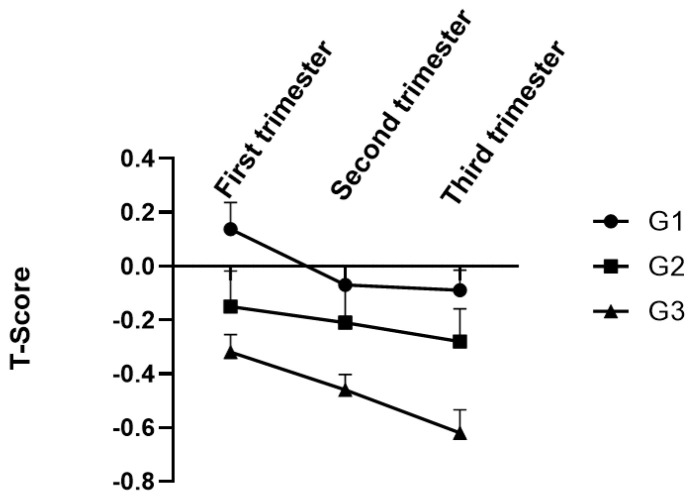
The changes in T-score during the whole pregnancy period across the tertiles of the DII for 289 pregnancy women.

**Table 1 nutrients-16-00455-t001:** The demographic characteristics across tertiles of DII in the baseline survey.

Characteristics	Tertiles of DII			*p* Value
	G1 (*n* = 96)	G2 (*n* = 96)	G3 (*n* = 97)	
Age, years	28.98	29.95	28.73	0.087
BMI, kg/m^2^	22.06	21.66	21.84	0.696
Educational level, *n*				
High school or below	56	45	55	0.071
College	32	38	28	
Postgraduate or above	6	11	10	
Time spent for physical activity, *n*				
<2 h/week	24	23	21	0.556
2–4 h/week	24	35	33	
4–6 h/week	24	18	26	
>6 h/week	22	18	13	
Nutrient intake				
Calcium, mg/day	1140	1143	1161	0.805
Vitamin D, IU/day	218.16	239.36	217.36	0.524
Isoflavone, mg/day	37.3	42.7	49.9	0.039
Time for sunshine, h/day	1.65	1.72	1.68	0.878

**Table 2 nutrients-16-00455-t002:** Distribution of food groups across tertiles of DII.

Food Groups	Tertiles of DII			*p* Value
	G1 (*n* = 96)	G2 (*n* = 96)	G3 (*n* = 97)	
Dietary energy, kcal/day	1957.74	2106.18	2310.87	
Refined Cereals, g/day	250.04	263.99	344.62	<0.001
Whole Cereals, g/day	11.97	23.95	39.99	<0.001
Soy and soy product, g/day	62.94	113.47	170.27	<0.001
Dark leafy vegetables, g/day	110.62	158.59	292.33	<0.001
Light vegetables, g/day	134.57	204.52	312.69	<0.001
Dark fruits, g/day	88.51	155.88	221.80	<0.001
Light fruits, g/day	116.08	152.61	189.10	<0.001
Poultry, g/day	9.18	18.02	24.77	<0.001
Meat and Processed meats, g/day	33.69	47.75	52.72	0.041
Fish and aquatic products, g/day	28.89	34.40	33.06	0.518
Eggs, g/day	44.68	61.69	52.44	0.013
Milk and Dairy products, g/day	208.31	221.71	270.17	0.046
Fungus and Alga, g/day	16.43	28.23	53.54	<0.001
Nuts, g/day	9.03	12.19	19.84	<0.001
Backed Bread, g/day	15.18	17.60	21.36	0.245
Candies, g/day	1.47	1.54	2.14	0.612
Junk food, g/day	1.70	1.31	1.19	0.724
Fruit and vegetable juice, mL/day	4.20	4.16	5.43	0.871
Soft drinks, mL/day	30.33	35.58	29.14	0.854

**Table 3 nutrients-16-00455-t003:** The assessment of serum inflammation factor levels across tertiles of DII.

Item	Tertiles of DII			*p* Value
	G1 (*n* = 96)	G2 (*n* = 96)	G3 (*n* = 97)	
CRP (mg/L)	3045.902	3104.019	3095.260	0.653
IL-6 (pg/mL)	2.019	2.474	2.862	0.034
IL-10 (pg/mL)	1.879	1.716	1.685	0.795
IL-4 (pg/mL)	1.956	1.925	1.806	0.879
IL-1β (pg/mL)	1.478	1.518	1.761	0.082

**Table 4 nutrients-16-00455-t004:** The assessment of maternal bone metabolism across the tertiles of DII.

Item	Tertiles of DII			*p* Value
	G1 (*n* = 96)	G2 (*n* = 96)	G3 (*n* = 97)	
OC	12.87	11.29	9.55	0.415
OPG	624.65	605.62	593.41	0.209
PINP	56.49	48.72	45.58	0.063
PTH	22.86	17.84	15.93	0.112
RANKL	5.34	7.89	10.43	0.002

**Table 5 nutrients-16-00455-t005:** The assessment of maternal BMD across the tertiles of the DII in the third trimester.

Item	Tertiles of DII			*p* Value
	G1 (*n* = 96)	G2 (*n* = 96)	G3 (*n* = 97)	
SOS	1399.65	1404.09	1403.26	0.978
Z-score	−0.11	−0.25	−0.44	0.694
T-score	−0.09	−0.28	−0.62	0.043

**Table 6 nutrients-16-00455-t006:** Adjusted means of maternal BMD changes across tertiles of DII.

Model Name	Tertiles of DII			*p* for Trend *
	G1 (*n* = 96)	G2 (*n* = 96)	G3 (*n* = 97)	
Model 1	0.03	0.01	−0.09	0.153
Model 2	−0.02	−0.07	−0.16	0.047

* A test for linear trend conducted by including the median value of each tertile as a continuous term in the regression analysis. Model 1: No adjustments were made. Model 2: Multivariate models were adjusted for demographic characteristics, namely, BMI, educational level, calcium and vitamin D consumption, and time spent on physical activity and in the sunshine.

## Data Availability

The data presented in this study are available on request from the corresponding author. The data are not publicly available due to the further article publication.

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
