# Peer review of "Association between Dietary Inflammatory Index and Bone Mineral Density Changes among Pregnant Women: A Prospective Study in China"

_nutrients, 2024, doi:10.3390/nu16030455_

Round 1

Reviewer 1 Report

Comments and Suggestions for Authors

In the study A prospective study of the association between dietary inflammatory index and bone density changes among pregnant women in China authors have measured bone density in preglant women and correleted it with dietary habits and proinflammatory markers. 

Minor comments:

- authors abbreviate tertiles (X/3) with Q, which is usually used for Quartillae (X/4). I suggest shortening with T1, T2 and T3, or group 1 (G1, G2, G3).

Text ln. 200-206 is inappropriately placed, and belongs either in the introduction or in the discussion.

In the presentation of the statistical result for T-score and Model 2, it is necessary to display the actual value for P= and not to write P< 0.05

Comments on the Quality of English Language

The study is well written, but due to numerous grammatical errors, it requires thorough language proofreading. I especially emphasize the improper use of capital letters for nouns within the text, and:

- ln. 33-34 one sentence is divided into two, so both are deficient.

- .. we also analyze the T-score

- ... DII tertiles which indicate that the

Minor comments:

- authors abbreviate tertiles (X/3) with Q, which is usually used for Quartillae (X/4). I suggest shortening with T1, T2 and T3, or group 1 (G1, G2, G3).

Author Response

Dear professor,

Thank you for your professional and detailed comments and suggestions. Those comments are valuable and very helpful. We have read through comments carefully and have made corrections. In response to your suggestions, we made a point-by-point revision of the article and delivered it to make a professional  language editing by officially recomended web site. Based on the instructions provided in your letter, we uploaded the file of the revised manuscript. The responses to comments are marked in red and presented following.

  1. Authors’ abbreviate tertiles (X/3) with Q, which is usually used for Quartillae (X/4). Reviewer suggested shortening with T1, T2 and T3, or group 1 (G1, G2, G3). So we changed all the tertiles (X/3) from Q to G1, G2 and G3.
  2. ln. 200-206 “Then, we analyzed the bone metabolism of the participants across DII tertiles. As shown, the RANKL level was significantly different across DII (P = 0.002) (Table 4) . Although the levels of OC, OPG, PINP and PTH were not significantly different across DII score, the decreasing trend was found with the increasing DII (P > 0.05). Meanwhile, the ratio of RANKL/OPG in the G3 was 0.017 which was the highest and lowest ration is 0.0085 in the G1. These data indicating that the DII score is associated with the pro-inflammation factor especially the IL-6 to affect the bone metabolism." is inappropriately placed, I reposition it in the discussion section (ln. 264-267).
  3. In the presentation of the statistical result for T-score and Model 2, it is necessary to display the actual value for P= and not to write P<0.05. So wedisplay the actualvalues which are 0.043 and 0.047 (ln. 225,244), respectively.
  4. All the grammatical errors that you pointed out for ushave been corrected.

We would love to thank you for allowing us to resubmit a revised copy of the manuscript and we highly appreciate your time and consideration.

Sincerely,

Xiaoyu Zhu

Reviewer 2 Report

Comments and Suggestions for Authors

The manuscript “A prospective study of the association between dietary inflammatory index and bone density changes among pregnant 3 women in China by Zhu et al.

The content is presented in a logical manner. The abstract provides a concise summary of the study's objectives, methods, key findings, and conclusions.

Here are some suggestions for improvement:

1.      Title can be improved. What do you think of this “Association of Dietary Inflammatory Index and Bone Density Changes in Pregnant Women: A Prospective Study in China”.

2.      Abstract: objectives can be improved. Here is a suggested revision: “This study aims to examine the relationship between dietary inflammatory index (DII) and bone density (BD) changes in Chinese pregnant women, offering valuable insights for dietary guidance during pregnancy.”

3.      Introduction;
Original: "Diet has been reported to play an important role in regulating human body inflammation level through pro or anti-inflammatory mechanisms, thus the concept of pro-inflammatory and anti-inflammatory diets has been widely reported recently[1,2]." Revised: "Diet plays a crucial role in regulating the body's inflammation levels through pro and anti-inflammatory mechanisms. Recently, there has been widespread reporting on the concepts of pro-inflammatory and anti-inflammatory diets[1,2]."

4.      Introduction:
Original: "While diet that contains more intakes of fruits, vegetables, whole grains, fish or green tea could help reduce inflammation, and was indicated as anti-inflammatory diets[5-7]." Revised: "Diets rich in fruits, vegetables, whole grains, fish, or green tea have been associated with reduced inflammation, categorizing them as anti-inflammatory[5-7]."

5.      Introduction:
Original: "Up to now, some dietary indices has been reported, trying to evaluate the total dietary quality." Revised: "To date, various dietary indices have been reported, attempting to evaluate overall dietary quality."

6.      Ethic committee: please include ID number and date of approval. Please also refer that the study follows Helsinki declaration.

7.      Additional limitations: Single-Center Study, small sample size. In addition: Bone Density Measurement: The use of ultrasonography for bone density measurement, while practical, may have limitations compared to more precise methods such as dual-energy X-ray absorptiometry (DXA). DXA is considered the gold standard for bone density assessment. Finally, the study design allows for the identification of associations, but it may not establish causation. Other factors, not explored in this study, could contribute to both dietary choices and bone density changes.

8.      The Mediterranean diet (MD) is widely acknowledged as one of the most effective dietary approaches for preventing non-communicable diseases. Emphasizing the consumption of vegetables, 1–2 portions of fruits per day, whole grains, extra virgin olive oil, legumes, nuts, and moderate amounts of fish and dairy products, while minimizing the intake of meat, processed, and industrial foods, the MD has demonstrated a positive impact on both oxidative stress and inflammation (Curr Issues Mol Biol. 2023 Aug 12;45(8):6651-6666. doi: 10.3390/cimb45080420). Could assessing the quality of food intake as a percentage, rather than focusing on individual dietary components, enhance our understanding of the association between bone health and inflammatory indices? Please discuss.

Comments on the Quality of English Language

Moderate revision

Author Response

Dear professor,

Thank you for your professional and detailed comments and suggestions. Those comments are valuable and very helpful. We have read through comments carefully and have made corrections. In response to your suggestions, we made a point-by-point revision of the article and delivered it to make a professional  language editing by officially recomended web site. Based on the instructions provided in your letter, we uploaded the file of the revised manuscript. The responses to comments are marked in yellow and presented following.

  1. We improvedthe title by your suggestion, which name is “Association between Dietary Inflammatory Index and Bone Density Changes among Pregnant Women: A Prospective Study in China”(ln. 2-4).
  2. We improved theobjectives, and the new revision is: “This study aims to examine the relationship between dietary inflammatory index (DII) and bone mineral density (BMD) changes among Chinese pregnant women, offering valuable insights for dietary guidance during pregnancy. ”(  10-12).
  3. We improved theintroduction, and the new revision is: “Diet plays a crucial role in regulating the body's inflammation levels through pro- and anti-inflammatory mechanisms. Recently, there has been widespread reporting on the concepts of pro-inflammatory and anti-inflammatory diets [1,2]."( 30-32).
  4. Changed original: "While diet that contains more intakes of fruits, vegetables, whole grains, fish or green tea could help reduce inflammation, and was indicated as anti-inflammatory diets[5-7]." to revised: "Diets rich in fruits, vegetables, whole grains, fish, or green tea have been associated with reduced inflammation, categorizing them as anti-inflammatory [5-7]. "(ln. 35-37).
  5. Changed original: "Up to now, some dietary indices has been reported, trying to evaluate the total dietary quality." to revised:"To date, various dietary indices have been reported, each attempting to evaluate overall dietary quality. "(ln. 37-38).
  6. We added ID number and date of approval forEthic committee (ln. 354-356).
  7. We added additional limitations:"First, thedietary data were collected based on a validated FFQ, in which recall bias could not be avoided, and self-reported dietary data have been known to suffer from measurement errors. Second, the use of ultrasonography for bone mineral density measurement may have limitations compared to more precise methods such as dual-energy X-ray absorptiometry (DXA). Third, only a few confounding factors were adjusted in the linear models, but other potential factors may also exist. Fourth, the cohort study design allows for the identification of associations, but it cannot establish causation, and more factors that could contribute to the DII and BMD need to be explored. Fifth, the study sample size was small, and it was only a single-center study, so larger cohorts and multicenter studies would also be necessary to validate the results." (ln. 326-335).
  8. The Mediterranean diet (MD) is widely acknowledged as one of the most effective dietary approaches for preventing non-communicable diseases. We discussed "Assessing the quality of food intake as a percentage, rather than focusing on individual dietary components, can enhance our understanding of the association between bone health and inflammatory indices"as follows: "An effective dietary approach can prevent non-communicable diseases. To evaluate a diet, we normally focus more on the quality of food intake as a percentage but not an individual component since the relationship between foods is complex and as interactions exist between different foods. Assessing overall diet could thus help us to understand the interrelationships between dietary components [40]."(ln. 252-256)

We would love to thank you for allowing us to resubmit a revised copy of the manuscript and we highly appreciate your time and consideration.

Sincerely,

Xiaoyu Zhu

Reviewer 3 Report

Comments and Suggestions for Authors

Dear Authors,

Your article titled "A prospective study of the association between dietary inflammatory index and bone density changes among pregnant women in China," focuses on analyzing the relationship between bone density (BD) and the Dietary Inflammatory Index (DII), which is significant from the perspective of preventing inflammatory bone diseases, including osteopenia and osteoporosis.

Here are my comments:

Minor comment:

Lack of explanations for abbreviations in the tables.

Major comments:

  1. 1. Lack of information on the impact of pregnancy on bone tissue and no references to studies, such as the work by Huang and Schooling from 2017, which assessed the association between inflammatory markers and bone mineral density.
  2. 2. There is a need to justify the search for pathological bone changes in young, pregnant women, especially considering that bone loss may begin in middle age, increasing the risk of osteoporosis later in life.
  3. 3. You should more precisely justify why a prolonged inflammatory state associated with a pro-inflammatory diet may affect bone mass in young women, particularly in the context of pregnancy.

Author Response

Dear professor,

Thank you for your professional and detailed comments and suggestions. Those comments are valuable and very helpful. We have read through comments carefully and have made corrections. In response to your suggestions, we made a point-by-point revision of the article and delivered it to make a professional  language editing by officially recomended web site. Based on the instructions provided in your letter, we uploaded the file of the revised manuscript. The responses to comments are marked in blue and presented following.

  1. We discussed impact of pregnancy on bone tissue as follows: "BMDchanges throughout a trimester due to the high calcium demand from the growth of the fetus and changes in maternal hormones. Decreased BMD has been more widely reported in pregnant women compared with non-pregnant women. Pregnancy can cause reversible bone loss, especially at trabecular sites "(ln 290-293). And showed the association between inflammatory markers and bone mineral density as follows: "The higher circulating levels of inflammatory markers also predict bone loss and resorption. For example, IL-6 is an inflammatory cytokine that is associated with bone loss and resorption. High levels of IL-6 were also reported to induce bone loss in individuals with osteoarthritis. Meanwhile, IL-6 and CRP inhibit the functions of osteoblasts, including the promotion of the proliferation, differentiation, and activation of osteoclasts, causing bone erosion and bone mass loss: " (ln 283-288).
  2. We addedthe search for pathological bone changes in young, pregnant womenmay increasing the risk of osteoporosis later in life. And made a dietary recommendation to prevent osteoporosis during pregnancy (ln.322-323,343-345)
  3. We justify why a prolonged inflammatory state associated with a pro-inflammatory diet may affect bone mass in young women, particularly in the context of pregnancy as follows: "DII has widely been reported to be associated with BMDin adults and older individuals, but there are currently few studies that have evaluated the association between the DII and bone metabolism and density among pregnant women in China. In our study, BMD was measured in participants using an ultrasonic method at the second and third trimesters. We found that participants with a higher DII score along with a higher serum level of IL-6 showed a high level of RANKL and a higher ratio of RANKL/OPG, indicating a risk of more bone mass loss. RANKL has been widely reported to be associated with bone mass loss. For example, Hu et al. showed that RANKL promotes osteoclast formation and bone loss. RANKL can be considered an early marker of bone mass loss for pregnant women, and inflammatory factors can interact with osteoblasts and osteoclasts to change the expression levels of RANK and RANKL, thus affecting BMD. Interventions targeting inflammatory levels can directly impact bone health. Here, we connected the association between RANKL, bone mass loss, inflammatory factors, and DII scores among pregnant women. A low-DII diet may lower the risk of bone loss among pregnant women " (ln 292-310).
  4. We made explanations for abbreviations in the tables(ln 114-119).

We would love to thank you for allowing us to resubmit a revised copy of the manuscript and we highly appreciate your time and consideration.

Sincerely,

Xiaoyu Zhu

Reviewer 4 Report

Comments and Suggestions for Authors

See the attached file. I suggest to replace BD by BMD Bone mineral density all across the manuscript. BMD is a more usual way to express the parameter discussed in this study.

Comments on the Quality of English Language

I took time to make some corrections but I cannot ensure that I came accross all mistakes. A last correction by a fluent English scientist is required.

Author Response

Dear professor,

Thank you for your professional and detailed comments and suggestions. Those comments are valuable and very helpful. We have read through comments carefully and have made corrections. In response to your suggestions, we made a point-by-point revision of the article. Based on the instructions provided in your letter, we uploaded the file of the revised manuscript. The responses to comments are marked in green and presented following.

  1. The article has made a professional language editing by officially recomended web site.
  2. All the grammatical errors that you pointed out for ushave been corrected.
  3. Micronutrients is not a mandatory, so we calculated VitD and Ca+ by the frequency of food intake and dietary supplements multiplied by their respective nutrient content coefficients. Andexplained as follows:"Calculations of vitamin D and calcium (from food and dietary supplements) consumption were conducted by referring to the US Department of Agriculture database" (ln. 136-138).
  4. We added in the discussion part concerning the lack of effect of soybean intake on the Bone Mineral Density in pregnant women as follows:"It needs to be pointed out that the consumption of soybean in G3 was higher than that in G1, but the maternal BMD level in G3 was lower than that in G1. Although it was known that soybean intake reduced inflammation levels and inhibited bone loss due to estrogenic isoflavones, the function here was not as potent as that observed in previous reports. This may be due tothe intrinsic estradiol plasma levels are huge in pregnant women during the 2nd and 3rd trimester, the low affinity of isoflavones on the estradiol receptors or the small sample size and single-center limitation that prevented us from observing a correlation." (ln 310-317) and showed the values of isoflavone in table one (ln. 180).
  5. We addedMean, median andSD or min/max values of DII for each tertile as follows: "The mean of the DII scores was 59.69±54.66 (mean±SD). The lowest DII score was -76.05, and the highest was 246.68, indicating that the DII values are very different between different individuals’ diets. We calculated the tertile of DII and divided all the participants into groups G1(7.13±15.97), G2(53.05±11.6), and G3(119.53±45.0) ordered from the lowest to the highest DII scores". (ln. 170-174).
  6. We definedQC, PINT and PTH, RANK-L, OPG abbreviations in the material and methodpart.(ln .118-120)
  7. We added complete identification of the tertiles DII values at the Material and Method part (ln. 142-148,158-159).
  8. Question: Can you please give the food consumption as aproportion or percentage of energy or mass intake. This would be better to explain where are themain dietary differences between Q1 and Q3 explaining the differences in DII. Answer: In Table 2, we listed the intake of various food groups. Considering the values are not energy intakes, it is not possible to calculate the percentage of energy consumed by each food group. So we added the explanation in the discussion part as follows: "The parameters of the DII mostly stemmed from certain food groups, such as refined cereal, which accounted for the largest proportion of the diets and might have led to the significant difference across the DII groups. The consumption of candies or junk foods indicated greater energy intake but affected the scores less. We hypothesize that this may because of the overall proportion of these foods is small compared to that of refined cereal". (ln. 274-279)

We would love to thank you for allowing us to resubmit a revised copy of the manuscript and we highly appreciate your time and consideration.

Sincerely,

Xiaoyu Zhu

Round 2

Reviewer 2 Report

Comments and Suggestions for Authors

Authors appropriately answered to all the issues I raised

Author Response

Dear professor,

Thank you for your letter and for the comments concerning our manuscript entitled “Association between Dietary Inflammatory Index and Bone Mineral Density Changes among Pregnant Women: A Prospective Study in China ”. We tried our best to improve the manuscript and made some changes in the manuscript.

We appreciate for your warm work earnestly, and your approval for accepting the revised manuscript for publication in Nutrients. Once again, thank you very much for your comments and suggestions.

Sincerely,

Xiaoyu Zhu

Reviewer 3 Report

Comments and Suggestions for Authors

Dear Professor Zhu,

I trust this message finds you well. Thank you for your prompt response and for diligently addressing my comments and suggestions on the manuscript. I have carefully reviewed the revised version along with your detailed responses, and I am pleased to acknowledge the significant improvements made.

Your comprehensive explanations regarding the impact of pregnancy on bone tissue, the association between inflammatory markers and bone mineral density, and the justification of a prolonged inflammatory state in young women, particularly during pregnancy, greatly enhance the clarity and depth of the manuscript. I appreciate the incorporation of the search for pathological bone changes in young, pregnant women and the dietary recommendations to prevent osteoporosis during pregnancy. The additional insights provided into the association between DII, RANKL, inflammatory factors, and bone mass loss among pregnant women add valuable depth to the discussion.

Furthermore, the explanations for abbreviations in the tables contribute to the overall clarity of the manuscript.

Considering the thorough revisions made, I am pleased to inform you that I accept the revised manuscript for publication in Nutrients. Your dedication to addressing the suggestions and making necessary corrections has resulted in a well-refined and comprehensive piece of work.

I would like to express my gratitude for your cooperation and diligence throughout this process. I commend your efforts in making the necessary adjustments and ensuring the manuscript's readiness for publication.

Please proceed with the necessary steps for submission, and I look forward to seeing the article published in Nutrients.

Sincerely,

Joanna Bartkowiak-Wieczorek

Author Response

(The authors gave the same response as above.)

Reviewer 4 Report

Comments and Suggestions for Authors

My comments are in the attached file.

Comments on the Quality of English Language

The English has been significantly improved. There are still a few mistakes remaining that would require careful editting.

Author Response

Dear professor,

Thank you for your letter and for the comments concerning our manuscript entitled “Association between Dietary Inflammatory Index and Bone Mineral Density Changes among Pregnant Women: A Prospective Study in China ”. Those comments are valuable and very helpful for revising and improving our paper, as well as the important guiding significance. We have studied comments carefully and have made corrections. Revised portion are marked in yellow in the paper. The main corrections in the paper and the responds to the reviewer’s comments are as following:

  1. Question: “The BMI values seem quite low especially if they were registered at the second and third trimesters.”Answer: The values in table one showed the baseline survey BMI of pregnant women. And the standard reference value of Chinese women of BMI is 18.5-24. Therefore, our results are in the range of 21.84-22.06, which are within the standard reference range.
  1. Question:“The values for isoflavones intake across the different tertiles seem much lower than what is classically reported in China where the mean isoflavones intake varies from 20 to 40 mg/day. Can you explain? How did you obtained these values? ” Answer: Thanks to this reminder, we have reconfirmed and updated the data.(Line 180) The isoflavones were examined with reference to the Hong Kong database. We calculated the values of isoflavones by the intake of soy, soy product and other cereals, like red beans, green beans. So the values are higher than the isoflavones from soy products:12.6; 22.7 and 34.1 mg/day. The updated values are 37.3; 42.7 and 49.9 mg/day, respectively.
  1. Question: Lines 304-312: Thank you for the comment on the lack of effect of isoflavones. However, another reason may be that it was shown that the active doses of isoflavones on the prevention of BMD losses are around 80 mg/day in an anti-inflammatory dietary context. See Corbi et al. (doi: 10.3390/ijms241512063) as an argument. Answer: Thank you for your comprehensive analysis and suggestions, we have added relevant content to the article.“This may be due to the relatively low amount intake of isoflavone here which was suggested to prevent BMD loss around 80 mg/d in postmenopausal women on a previous study [53]. Meanwhile, the health benefits of isoflavones is very depend on the equol, the ability of a individual to produce equol is very different which could also affect the benefit of isoflavones.” (Line 314-316, 319-321) And cited relevant reference at the same time.
  1. Line 325, repalced“liner” with “linear”
  2. Line 341, repalced “isnecessary” with “are necessary”

We appreciate for your warm work earnestly, and hope that the correction will meet with approval. Once again, thank you very much for your coments and suggestions.

Sincerely,

Xiaoyu Zhu